# Maternal complications in pregnancy and childbirth for women with epilepsy: Time trends in a nationwide cohort

Kim Christian Danielsson[1,2]*, Nils Erik Gilhus[2,3], Ingrid Borthen[2], Rolv Terje Lie[4,5], Nils-Halvdan Morken[1,6]

1 Department of Obstetrics and Gynecology, Haukeland University Hospital, Bergen, Norway, 2 Department of Clinical Medicine, University of Bergen, Bergen, Norway, 3 Department of Neurology, Haukeland University Hospital, Bergen, Norway, 4 Department of Global Public Health and Primary Care, University of Bergen, Bergen, Norway, 5 Centre for Fertility and Health, Norwegian Institute of Public Health, Oslo, Norway, 6 Department of Clinical Science, University of Bergen, Bergen, Norway

* kim.christian.danielsson@helse-bergen.no

**Data Availability Statement:** All data from the Medical Birth Registry of Norway can be accessed via application to Norwegian Institute of Public Health. Requirements for application and a link to

## Abstract

### Objective

Obstetric trends show changes in complication rates and maternal characteristics such as caesarean section, induced labour, and maternal age. To what degree such general time trends and changing patterns of antiepileptic drug use influence pregnancies of women with epilepsy (WWE) is unknown. Our aim was to describe changes in maternal characteristics and obstetric complications in WWE over time, and to assess changes in complication risks in WWE relative to women without epilepsy.

### Methods

This was a nationwide cohort study of all first births in the Medical Birth Registry of Norway, 1999–2016. We estimated maternal characteristics, complication rates, and risks for WWE compared to women without epilepsy. Main maternal outcome measures were hypertensive disorders, bleeding in pregnancy, induction of labour, caesarean section, postpartum hemorrhage, preterm birth, small for gestational age, and epidural analgesia. Time trends were analyzed by logistic regression and comparisons made with interaction analyses.

### Results

426 347 first births were analyzed, and 3077 (0.7%) women had epilepsy. In WWE there was an increase in proportions of induced labour (p<0.005) and use of epidural analgesia (p<0.005), and a reduction in mild preeclampsia (p = 0.006). However, the risk of these outcomes did not change over time. Only the risk of severe preeclampsia increased significantly over time relative to women without epilepsy (p = 0.006). In WWE, folic acid supplementation increased significantly over time (p<0.005), and there was a decrease in smoking during pregnancy (p<0.005), but these changes were less pronounced than for women without epilepsy (p<0.005).

the electronic application are available at: https://www.fhi.no/en/op/data-access-from-health-registries-health-studies-and-biobanks/medical-birth-registry-and-registry-of-pregnancy-termination/access-to-data/

**Funding:** The authors received no specific funding for this work.

**Competing interests:** The authors have declared that no competeing interests exist.

## Conclusions

During 1999–2016 there were important changes in maternal characteristics and complication rates among WWE. However, outcome risks for WWE relative to women without epilepsy did not change despite changes in antiepileptic drug use patterns. The relative risk of severe preeclampsia increased in women with epilepsy.

## Introduction

Epilepsy is one of the most common chronic diseases during pregnancy.[1–4] Women with epilepsy (WWE) have been considered as high risk parturients with increased risk for maternal complications.[2–8] Almost half of women with ongoing or previous epilepsy use antiepileptic drugs (AEDs) in pregnancy to control seizures despite their potential adverse effects on the fetus and maternal complications.[2, 9–11] The pattern of antiepileptic drug use in pregnant WWE has changed markedly during the last two decades owing to newer antiepileptic drugs, primarily lamotrigine and levetiracetam, replacing older antiepileptic drugs, such as carbamazepine, phenytoin, and valproate. [12–14] The newer antiepileptic drugs are better tolerated and believed to have less fetal and maternal adverse effects, but are associated with increased seizure risk during pregnancy.[10, 11, 15–19] Increasing maternal age, increasing maternal body mass index (BMI), and decrease in smoking during pregnancy over the last two decades should also affect WWE.[20–23] These factors could be proportional or have a more complex interaction. Global trends show an increase in caesarean section rates and increased induction of labour.[24–26] Such interventions are common in WWE.[2, 4, 5, 7, 8] During the last decade, there has been an increasing focus on management of WWE during pregnancy and delivery and recent guidelines encourage close monitoring of pregnancies in WWE and strict indications for interventions.[25, 27–30] However, there is little data on how focused management and guidelines have affected maternal outcomes of WWE. A recent meta-analysis indicates a trend towards increasing rates of caesarean section and induction of labour in WWE. [31] However, different geographical populations with great variation in obstetric practice were compared to describe differences over time, and no reference populations were included. Therefore, it is not known how changes in population characteristics, obstetric practice and general complication rates have affected WWE. We expect that changes during the recent years in folate use, indications for operative interventions, and AEDs used have all influenced maternal complications in WWE during pregnancy and when giving birth.

By analyzing a stable nationwide cohort over 18 years, our aim was to describe changes in maternal characteristics and maternal complication rates in WWE over time, and to assess changes in complication risks relative to women without epilepsy. For changes in outcome risks in WWE over time, the influence of AED use and other specific factors were assessed.

## Material and methods

This population-based, nationwide study used data from the Medical Birth Registry of Norway (MBRN) from 1999 through 2016. Reporting to the MBRN is compulsory from all maternity wards. All pregnancies from 12 weeks gestation are recorded. Data are prospectively registered throughout pregnancy, delivery and postpartum period on a standardized form that has been unchanged since 1999. All data are forwarded to the MBRN by the attending midwife or obstetrician. Data includes maternal and paternal social factors, maternal health prior to and during

pregnancy, complications and interventions during pregnancy and birth, perinatal outcome, and complications in the children. The MBRN is routinely linked to the National Population Database via a unique social security number, given to all citizens in Norway, to ensure notification of all births in the country.[32]

Only first pregnancies were included and analyzed, excluding subsequent pregnancies from the same women in order not to interfere with or dilute outcomes.[33] Only singleton pregnancies ≥ 22 weeks of gestation were included, thereby excluding multiple gestations (1.8%). Women with missing social security number were also excluded (0.6%), representing foreigners or immigrants without permanent residence.

Epilepsy was registered in a checkbox at the standardized form. Information on type of epilepsy or seizure activity was not registered. The epilepsy diagnosis in MBRN has previously been found to be correct in 92.3% of cases.[9] Information on AED use in pregnancy (identified by ATC classification) was recorded as written text with no information on dose or therapeutic changes throughout pregnancy. Registration of AED use in MBRN has previously been found to be correct in 91.7% of cases.[33, 34] We identified and analyzed any AED use (all AED use including both mono- and polytherapy treatment). We analyzed also all WWE without any AEDs during their pregnancy. Furthermore, an a priori decision was made to analyze the four most common AEDs in monotherapy. This was performed to obtain both homogenous and large enough groups.

Time was examined as an effect measure modifier. The exposure of time was stratified according to year of birth. To obtain large enough groups and to increase rare outcomes in specified time categories, three subsequent years were merged into a total of six triennial time categories; 1999–2001, 2002–2004, 2005–2007, 2008–2010, 2011–2013, and 2014–2016.

Main outcomes were; a compound variable of any hypertensive disorder, mild and severe preeclampsia, bleeding in pregnancy, induction of labour at term, emergency and elective caesarean sections, postpartum hemorrhage, spontaneous preterm birth, small for gestational age, and epidural analgesia. Spontaneous preterm birth, < 37 weeks, (induction of labour and preterm elective caesarean sections were excluded from these analyses) was calculated from ultrasound assessment of due date. Last menstrual period was used for gestational age estimation in 2.3% of all births where ultrasound dates were missing. Mild preeclampsia was defined as persisting blood pressure ≥140/90 mmHg combined with proteinuria ≥ 0,3g per 24 hours. Severe preeclampsia was defined as blood pressure ≥160/110mmHg, proteinuria ≥3 g per 24 hours, oliguria or clinical symptoms of preeclampsia. Severe preeclampsia also included all cases of early onset preeclampsia (<34 weeks), eclampsia, or HELLP syndrome (hemolysis, elevated liver enzymes, low platelets). The compound variable of any hypertensive disorder was defined as the presence of any of these hypertensive disorders or gestational hypertension. Superimposed preeclampsia in women with pre-gestational hypertension was not specified but was included in the respective subcategories of hypertensive complications by severity. Bleeding in pregnancy included any vaginal bleeding throughout pregnancy reported by health workers in the primary care or in hospital. Induction of labour included induction at term (37–41 weeks), regardless of method. Caesarean section was reported as either an elective or an emergency procedure. Bleeding during delivery and within the next 24 hours after delivery (postpartum hemorrhage) was reported as > 1500ml, and all caesarean deliveries were excluded. Small for gestational age was defined as sex specific birthweight below the 2.5 percentile by gestational age, using z-score of all first births. Epidural analgesia was registered as any use throughout delivery. Other epilepsy core maternal outcomes suggested by the CROWN initiative were either not properly addressed in the MBRN or severely underpowered for inclusion in our study.[35] All outcomes were registered in checkboxes in the MBRN, except gestational age and birthweight.

Other variables were: mean maternal age, maternal employment status (employed or not), maternal nationality (native Norwegian or other), marital status (married or cohabitant vs. single), maternal pre-existing chronic disease (kidney disease, hypertension, diabetes mellitus), early gestational BMI, any self-reported smoking during pregnancy, use of folic acid supplementation in pregnancy (nationally recommended standard dose 0,4mg/day), and maternal hospital stay in days. BMI was only available from the year 2006 and onwards.

### Statistical analyses

We estimated total risk of all outcomes for WWE compared to women without epilepsy. Risks of outcomes were estimated in WWE treatment subgroups i.e. all WWE, WWE without AEDs, WWE with AEDs (any treatment in both mono- and polytherapy was included), and WWE with the four most common AEDs in monotherapy.

Logistic regression was used to estimate odds ratios (ORs) and interaction with adjustment for possible confounding by maternal age, smoking, folic acid supplementation and chronic diseases (pre-existing hypertension, kidney disease, and diabetes mellitus) for all outcomes. Adjustment by induction of labour and caesarean section was made where appropriate. Placenta previa, vaginal bleeding in pregnancy, augmented contractions, maternal nationality, and operative vaginal delivery were considered as possible confounders but not included in the final models, as they did not change the estimates.

Adjusted average effects for the whole study period of having epilepsy on outcomes without the effect of time were calculated and presented in tables. Time trends were then analyzed separately for WWE and women without epilepsy and interaction analyses were then used to compare risk changes over time in WWE relative to women without epilepsy for maternal characteristics and risk of outcomes. Thereafter, adjusted time trends and interactions were estimated. The time varying effect on proportions and risk changes over time were further explored in stratification-based sensitivity analyses. This was performed by analyzing time trends and interactions repeatedly in strata of AEDs and covariates. To probe the interaction in analyses, independently estimated adjusted ORs (aOR) for each triennial time category was presented as graphs. In our analysis we put emphasis on estimates and their confidence intervals. P-values are used to test for trends and differences in trends between WWE and women without epilepsy.

All analyses were performed with IBM SPSS (Statistical Package for Social Sciences) version 24.0.

The study was approved by the Regional Ethics Committee of Western Norway (REK 2013/186). Use of Norwegian national register data does not require consent from participants.

## Results

A total of 426 347 women gave birth for the first time 1999–2016 (41.6% of all births within the time period), of whom 3077 (0.7%) were WWE. The proportion of WWE was stable (0.7–0.8%) until 2011, with a decrease to 0.5% thereafter (p <0.005). AEDs were used by 1200 (39.9%) WWE, and 1022 of them were on monotherapy during pregnancy. The total number of WWE on AEDs was unchanged during the study period (p = 0.125). The four most common AEDs in monotherapy were lamotrigine (N = 437), carbamazepine (N = 243), valproate (N = 130), and levetiracetam (N = 118). The pattern of AED use changed over time with increasing use of lamotrigine and levetiracetam while the use of carbamazepine and valproate decreased (p<0.005 for all) (S1 Fig).

In time trend analyses the proportion of induction of labour in WWE had the most noticeable increase with a doubling from 11.9% in 1999–2002 to 26.7% in 2014–2016 (p<0.005) (Fig

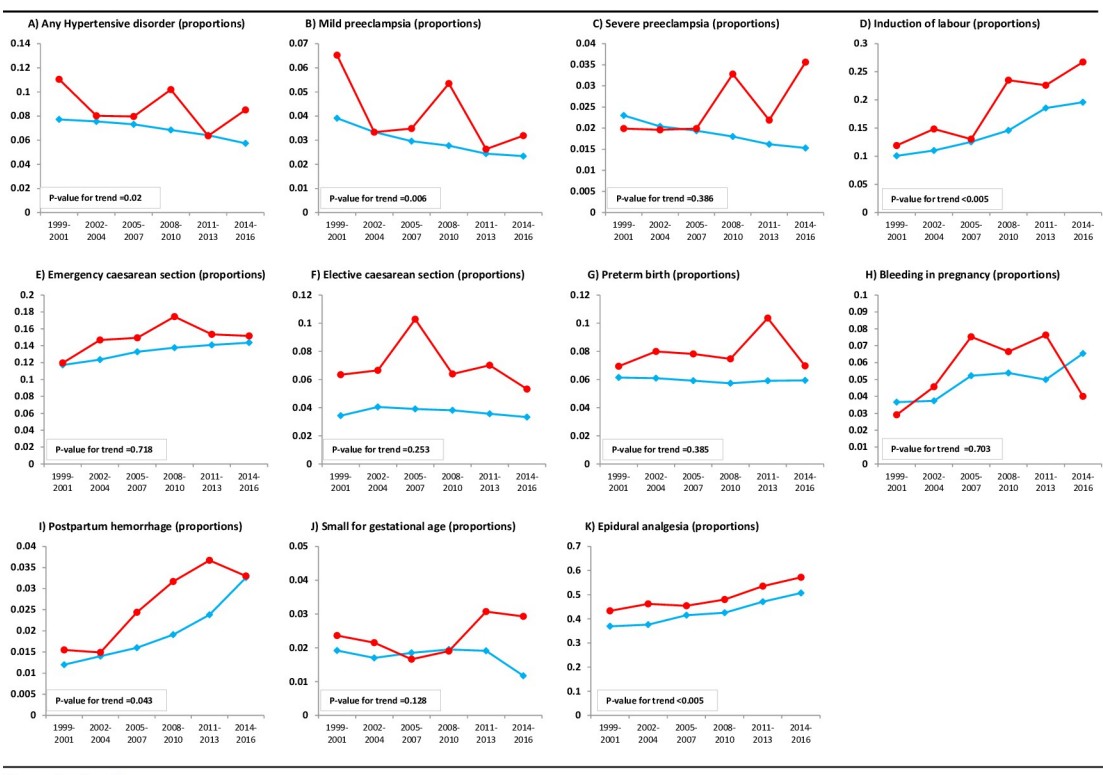

**Fig 1. Time trends for complications in women with epilepsy and women without epilepsy for first births, 1999–2016.**

1). There was also an increase over time in the proportion of WWE with epidural analgesia (p<0.005) and postpartum hemorrhage (p = 0.043). The proportion of any hypertensive complications and mild preeclampsia decreased over time in WWE (p = 0.02 and P = 0.006 respectively) as well as in the subgroup on AEDs.

Time trends in WWE were examined relative to time trends in women without epilepsy to estimate change in risk of outcomes. The proportion of severe preeclampsia in WWE changed from 2.0% to 3.6% during the study period. This increase was not significant (p = 0.39). In contrast, for the reference population without epilepsy the proportion of severe preeclampsia was reduced from 2.3% to 1.5% (p<0.005). Consequently, the relative risk for severe preeclampsia in WWE compared to women without epilepsy increased significantly over time (interaction p = 0.006) (Fig 2).

In sensitivity analyses, this increase in relative risk for WWE did not change in parallel to changes in any AED use or other covariates. Only when a combination of risk factors were included (high maternal age, unemployed, single parent, smoking, and other chronic diseases), the increase in relative risk of severe preeclampsia in WWE was reversed. Thus, the relative increase in risk of severe preeclampsia in WWE was influenced by factors associated with socioeconomic status. Fig 2 shows the aORs for all outcomes by three year periods (triennials) with corresponding logistic regression interaction analyses for WWE compared to women without epilepsy. Although there were significant changes in complications for both WWE and women without epilepsy during the examined time period, the relative risk for WWE

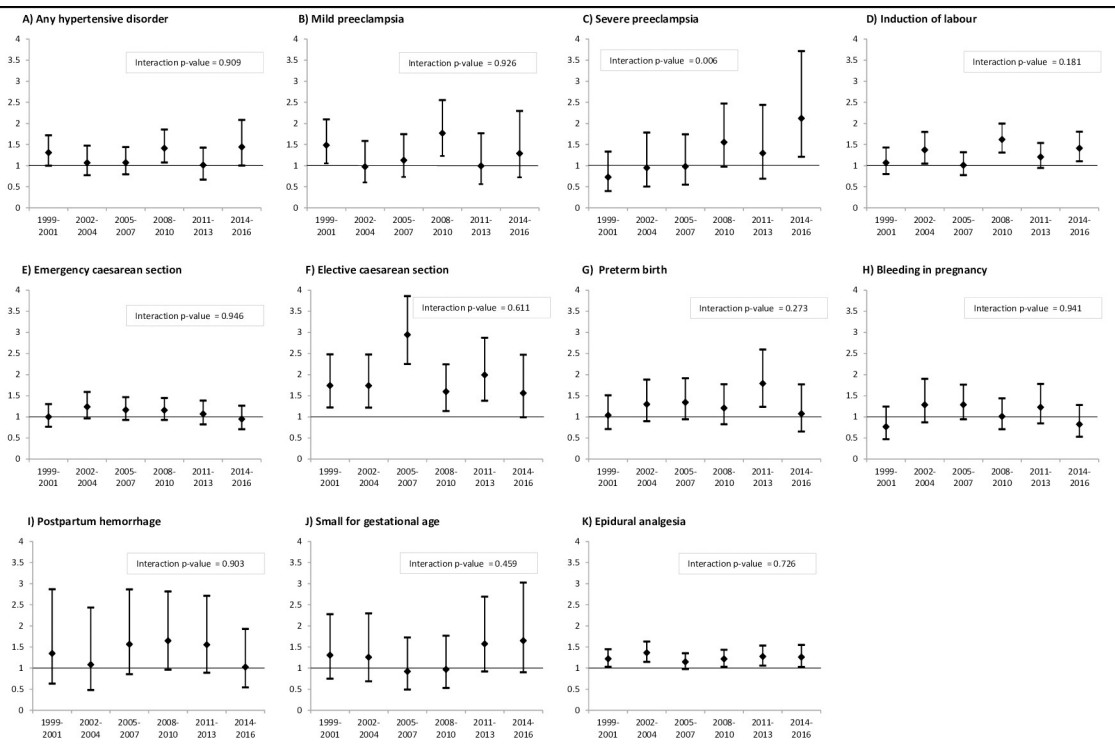

Odds ratios are merged for three year triennials with corresponding 95 % confidence interval.
The given interaction p-values represent risk change over time for women with epilepsy relative to women without epilepsy.

**Fig 2. Time trends for risks in women with epilepsy relative to women without epilepsy for first births, 1999–2016.**

did not differ over time for the remaining parameters; any hypertensive disorder, mild pre-eclampsia, induction of labour, emergency or elective caesarean section, preterm birth, bleeding in pregnancy, postpartum hemorrhage, small for gestational age, and use of epidural analgesia.

For WWE with lamotrigine monotherapy there was a borderline-significant reduction in relative risk of elective caesarean sections compared to the reference population during the study (interaction p = 0.06). WWE with lamotrigine also had a significant trend for increase in proportion of induction of labour (p = 0.008). Relative risk estimates for WWE did not change during the study period when changes in AED monotherapies or changes in other covariates were included in the sensitivity analyses. The outcomes for the total study period for WWE with and without AEDs are shown in Table 1, and for the four most common AEDs in mono-therapy in S1 Table. WWE had increased risk for any hypertensive disorder (aOR = 1.24, 95% CI: 1.09–1.41), mild preeclampsia (aOR = 1.35, 95% CI: 1.13–1.61), induction of labour (aOR = 1.22, 95% CI: 1.10–1.35), elective caesarean section (aOR = 1.99, 95% CI: 1.73–2.29), spontaneous preterm birth (aOR = 1.26, 95% CI: 1.08–1.47), and use of epidural analgesia (aOR = 1.20, 95% CI: 1.12–1.29) compared to women without epilepsy. Induction of labour was increased in WWE with lamotrigine (aOR = 1.55, 95% CI: 1.22–1.98) and levetiracetam (aOR = 2.67, 95% CI: 1.78–4.02). All four AEDs were associated with increased risk of elective caesarean section and epidural analgesia.

Maternal characteristics for the total study period are shown in S2 Table. There were significant changes in population characteristics in WWE during the registration period (Fig 3). Time trends for maternal characteristics showed a lower decrease in proportion of smoking

**Table 1. Total complications in first pregnancies of women with epilepsy (WWE) with and without antiepileptic drugs (AED) compared to women without epilepsy 1999–2016.**

| | Women without epilepsy | WWE | | WWE without AED | | WWE with AED | |
|---|---|---|---|---|---|---|---|
| | 423 270 | 3077 | | 1877 | | 1200 | |
| | N (%) | N (%) | aOR | N (%) | aOR | N (%) | aOR |
| **Any hypertensive disorder** | 29 231 (6.9) | 270 (8.8) | 1.24 (1.09–1.41) | 160 (8.5) | 1.18 (1.00–1.39) | 110 (9.2) | 1.33 (1.09–1.62) |
| **Mild preeclampsia** | 12 445 (2.9) | 129 (4.2) | 1.35 (1.13–1.61) | 73 (3.9) | 1.22 (0.97–1.55) | 56 (4.7) | 1.55 (1.18–2.03) |
| **Severe preeclampsia** | 7875 (1.9) | 75 (2.4) | 1.20 (0.95–1.52) | 51 (2.7) | 1.30 (0.98–1.72) | 24 (2.0) | 1.04 (0.69–1.56) |
| **Induction of labour** | 53 203 (14.6) | 477 (18.3) | 1.22 (1.10–1.35) | 239 (15.2) | 0.98 (0.85–1.13) | 238 (23.2) | 1.61 (1.39–1.86) |
| **Emergency cesarean section**\* | 56 307 (13.3) | 459 (14.9) | 1.09 (0.99–1.21) | 266 (14.2) | 1.05 (0.92–1.20) | 193 (16.1) | 1.16 (0.99–1.36) |
| **Elective cesarean section** | 15 572 (3.7) | 220 (7.1) | 1.99 (1.73–2.29) | 127 (6.8) | 1.86 (1.55–2.24) | 93 (7.8) | 2.19 (1.76–2.71) |
| **Preterm birth** | 19 725 (6.0) | 173 (7.9) | 1.26 (1.08–1.47) | 116 (8.3) | 1.28 (1.06–1.55) | 57 (7.2) | 1.21 (0.92–1.59) |
| **Bleeding in pregnancy** | 20 989 (5.0) | 171 (5.6) | 1.06 (0.91–1.24) | 106 (5.6) | 1.15 (0.94–1.40) | 65 (5.4) | 0.94 (0.73–1.21) |
| **Postpartum hemorrhage** | 6963 (2.0) | 61 (2.5) | 1.29 (0.99–1.67) | 32 (2.2) | 1.12 (0.79–1.60) | 29 (3.2) | 1.54 (1.06–2.23) |
| **Small for gestational age** | 7788 (1.8) | 70 (2.3) | 1.23 (0.97–1.56) | 39 (2.1) | 1.09 (0.80–1.51) | 31 (2.6) | 1.47 (1.03–2.10) |
| **Epidural analgesia** | 181 531 (42.9) | 1486 (48.3) | 1.20 (1.12–1.29) | 813 (43.3) | 1.00 (0.92–1.12) | 673 (56.1) | 1.59 (1.42–1.79) |

aOR = Adjusted odds ratio.

All outcomes adjusted for: maternal age, smoking, folic acid supplementation, chronic diseases.

\*Also adjusted for induction of labour.

during pregnancy for WWE (24.3% to 10.9%) than for women without epilepsy (22.2% to 5.3%), thereby increasing the relative difference over time (interaction p<0.005). The use of folic acid supplementation was more common in WWE than in women without epilepsy. The proportion of supplementation increased over time in both groups (p<0.005), but the increase was more pronounced in women without epilepsy (interaction p< 0.005). At the end of the study period, folic acid use was equally common in both groups. The mean maternal age for first births in WWE increased from 26.9 to 28.6 years from the first to the last triennial. This increase did not differ between groups (interaction p = 0.096). Marriage rates were lower for WWE than for women without epilepsy, and this difference increased during the study period (interaction p = 0.041). Among WWE, there was also an increase in use of assisted reproductive technology while there was a reduction in mean birth weight, mean length of maternal hospital stay and the proportion of native Norwegians.

Preterm births were in this study defined to include only spontaneous preterm births. Therefore we analyzed separately all preterm births, including also those with induction of labour and elective caesarean section (240 births in WWE and 26 139 births in women without epilepsy). Induction of labour and elective cesarean are indicated more frequently in pregnancies with various types of pathology. Our sub-analyses explored such associations. In all preterm births, WWE had an increasing proportion of severe preeclampsia from 11.6% in the first time period to 34.6% in the last period (p = 0.01). During the study period the risk of severe preeclampsia increased in WWE with preterm births relative to women without epilepsy with preterm births (interaction p<0.005). In WWE with preterm births, there was no increased risk for any other complications compared to preterm births in women without epilepsy.

BMI was only registered from 2006 and it was therefore not possible to account for this variable across the complete study period or to properly estimate the effect of BMI on changes over time. The BMI variable was also strained by missing values in 57% of cases after 2006. Missing values did not differ between WWE and women without epilepsy. We did not find any change in mean BMI 2006–2016, neither in WWE nor in women without epilepsy.

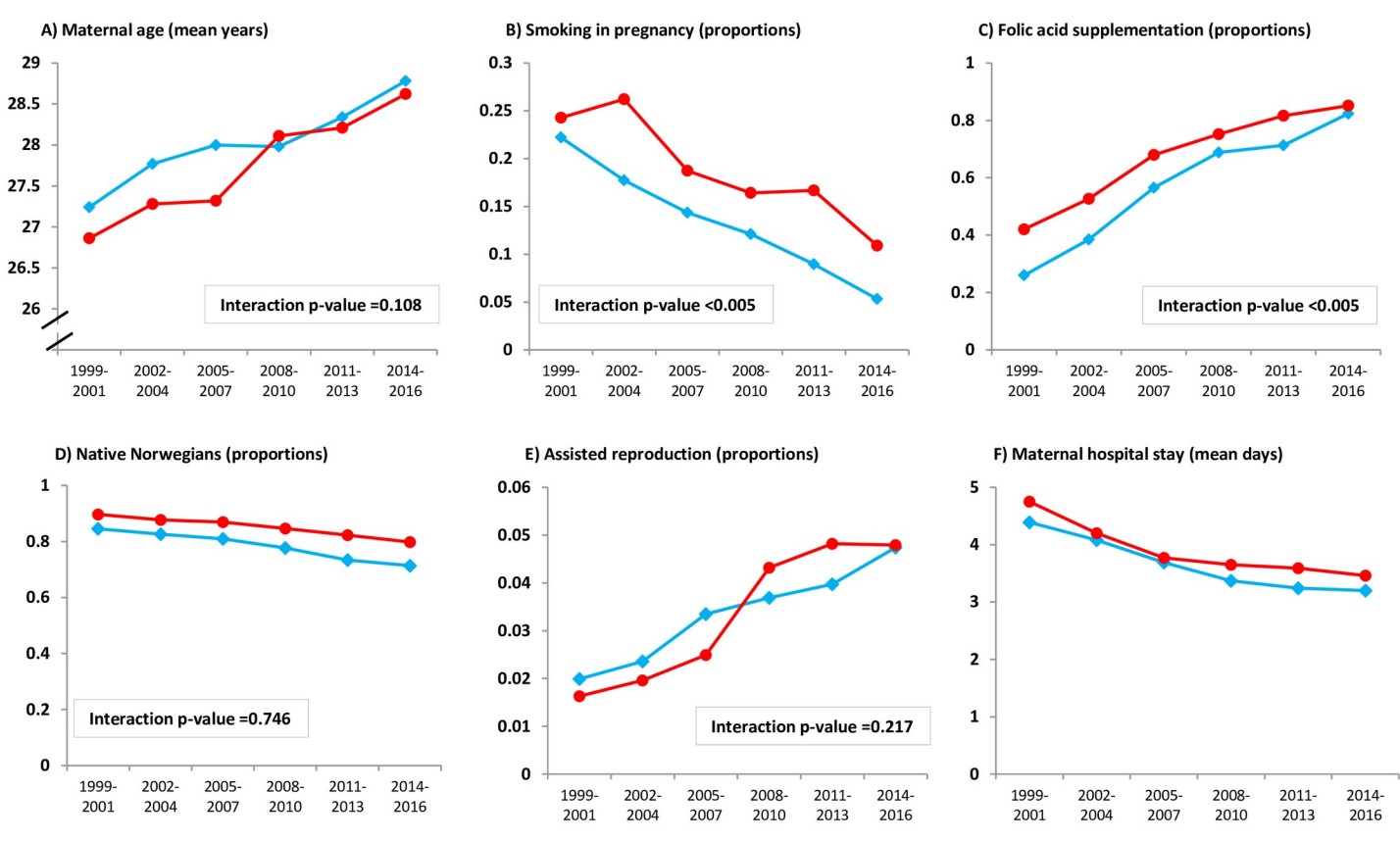

Women with epilepsy ●
Women without epilepsy ◆
Proportions and values are merged for three year triennials.
The given p-values for trend represent change over time for women with epilepsy.
Interaction p-values represent change over time in women with epilepsy relative to women without epilepsy.
For women with epilepsy, p-values for trend were <0.005 for all characteristics. This was true also for women without epilepsy.

**Fig 3. Time trends for maternal characteristics at first birth 1999–2016 in women with epilepsy and women without epilepsy.**

## Discussion

The relative risk for severe preeclampsia increased in WWE from 1999 to 2016. However, the proportion of severe preeclampsia in WWE did not change significantly during the study period. The increased risk was attributed to a reduction in the reference population, women without epilepsy. Although there was an increase in induction of labour and reduction in mild preeclampsia in WWE, the risks of maternal complications did not change relatively over time. This was regardless of increasing obstetric awareness of WWE during the study period and fundamental changes in the pattern of AED use.

We report pronounced changes during the most recent years in maternal characteristics and complication rates by a detailed examination of a complete, nationwide epilepsy population, and also changes relative to those occurring in women without epilepsy. WWE represent only 0.3–0.7% of all births.[1–4] Accordingly, large study samples and wide temporal intervals are necessary to enable estimates of infrequent complications. In our study, we present a temporal interval of 18 years. During this period, the occurrence of several maternal characteristics and complications varied more over time among WWE than between the two compared

groups, WWE and women without epilepsy (Fig 3). Accordingly, accurate present-day frequencies differed markedly from the corresponding average for the total study period (presented in S2 Table). Although not all temporal changes were significant in interaction analyses, additive changes appeared in such analyses. Therefore, temporal changes of characteristics, outcomes and medication use are notable when estimating risks in WWE.

We report a concerning increase in risk of severe preeclampsia in WWE relative to women without epilepsy. The increasing relative risk was especially evident in WWE with preterm births. AED use and other specified covariates in our study could not explain this increase. However, a combination of risk factors was important for the development of severe preeclampsia in WWE. This is in accordance with a recent study on hypertensive pregnancy complications in WWE.[34] Our findings emphasize a need for focused efforts on modifiable risk factors, lifestyle and better health care for WWE to reduce the risk of severe preeclampsia.

Proportions with mild preeclampsia decreased in WWE during the study period. Mild preeclampsia accounts for one-half of all hypertensive disorders, so the proportion of any hypertensive disorder also declined. There is an apparent controversy in our findings as mild preeclampsia became less common while the risk of severe preeclampsia increased over time in WWE. However, mild and severer forms of preeclampsia are suggested to have different pathophysiology with different factors influencing their development.[36] Therefore, factors influencing mild and severe preeclampsia can develop differently over time. In women without epilepsy, both mild and severe preeclampsia were reduced over time. Thus, the increase in risk of severe preeclampsia is due to an epilepsy-specific effect. Proportions of induced labour increased in WWE 1999–2016. However, the relative risk did not increase over time compared to women without epilepsy. Thus, this increase does not represent an epilepsy-specific trend. A previous meta-analysis studying maternal complications in WWE presented temporal changes by comparing different geographical populations.[31] Without any reference populations, they reported an increase over time in caesarean section rates and induction of labour for WWE. We found elective caesarean section to be more frequent in WWE than in women without epilepsy, but without any increase over time. Nor was there any increase in proportion of emergency caesarean section in WWE. In contrast, this proportion increased in women without epilepsy but the difference in trends was not significant. A survey by the World Health Organization has previously found increasing elective caesarean sections in first births as the most important contributor to the increase in cesarean section rates in high-income countries.[37] In our study, the proportion of elective caesarean sections declined in women without epilepsy. Compared to other European countries Norway has a low total caesarean section frequency.[22, 23] This applies also for WWE as the proportion of caesarean section was considerably lower than previously reported for this group.[4, 7, 31] Despite our low baseline proportion, WWE with AEDs had a trend for a further decrease of elective caesarean section, in particular for WWE with lamotrigine treatment. WWE with lamotrigine at the same time had an increase in induction of labour. The World Health Organization recommends improved case selection for induction of labour to reduce caesarean section rates.[37] During the study period, there has been an increase in focus on management of WWE during pregnancy and delivery in Norway illustrated by the numbers of studies, guidelines and consensus reports. [2, 6, 8, 25, 27, 28, 30] Lamotrigine as monotherapy has increasingly been the preferred drug for WWE with well-controlled epilepsy.[38] WWE on lamotrigine have at the same time more often been selected to induction of labour instead of a primary caesarean section. These changes for WWE possibly indicate improved preconceptional management, as suggested by these guidelines. We did not have any numerical measures or surrogate variables to assess effects of an assumed professional development in the management of WWE during the study period.

The proportion of WWE that smoked during pregnancy was halved. However, the decrease among women without epilepsy was even greater, consequently increasing the relative difference at the end of the study period. Use of folic acid supplements among WWE increased from 42% to 85% during the study period. However, the increase was even more pronounced in women without epilepsy. These changes imply positive health effects for WWE over time, but also room for improvement. Preconceptional counseling and focused personalized management is important for all WWE. The updated international EURAP cohorts of women with epilepsy giving birth shows a similar increase in use of folic acid over time, but with considerably lower absolute numbers, and therefore even more room for improvement.[39, 40]

WWE had longer hospital stays both after operative and vaginal deliveries compared to women without epilepsy. The reduction from 4.7 to 3.5 days in WWE during the time period was proportional to that seen in women without epilepsy, showing a persistent need for extra postpartum care in WWE.

Strengths of our study are a complete nation-wide registration of all births in a large and homogenous population. Exclusion of subsequent pregnancies ensures less dilution and interference on outcomes, as outcome risks in later pregnancies change significantly, also in WWE.[33] The diagnosis of epilepsy and AED use in the MBRN during pregnancy are reliable variables.[9, 33, 34] The standardized and unchanged registration, confirmed by the consistent time trends for the reference population, makes certain that the temporal changes are true rather than internally administrative. Any changes in coding practice would be expected to affect WWE and the reference population to the same degree. Outcomes registered in MBRN are precisely defined enabling a detailed exploration. An ascertainment bias for epilepsy as a chronic disease cannot be excluded. An ascertainment bias would entail more thorough registration in WWE possibly increasing some unfavorable outcomes. However, precise definitions and national guidelines should reduce this possibility.[25] Furthermore, several variables in the MBRN have been validated with acceptable to very good results. [34, 41–47] The total outcomes for WWE in our study are in line with previous findings, indicating generalizability of our newly explored time trends.[2, 4, 5, 7, 8] Being population based, the MBRN data does not suffer from referral bias. However, WWE are at risk of early pregnancy terminations, consequently possibly excluding some women with severe epilepsies from the actual birth cohort.[4, 48] As MBRN does not register early pregnancies, we could not address this possible selection. An important limitation of our study is the lack of data on seizure activity and type of epilepsy. Type of epilepsy and seizures during pregnancy has previously not been associated with maternal pregnancy complications, and epilepsy types are not expected to have changed 1999–2016.[9, 49] However, AED prescription patterns changed fundamentally during the study. First line treatment for women in fertile age is now more frequently lamotrigine and levetiracetam. These newer AEDs are associated with higher seizure frequencies in pregnancy.[17–19] Females still treated with valproate and carbamazepine in the last years of the study period may have had more severe and treatment-resistant epilepsies. Data on adverse effects of valproate in pregnancy increased during the study period, culminating in the strengthened warning by the European Medicines Agency in 2014 on the use of valproate in women of reproductive age.[50] WWE with AED monotherapy had low numbers of most complications, and therefore our study lacks strength for some variables despite the large cohort. BMI is an important covariate when exploring obstetric complications. The BMI variable in the MBRN is strained in two dimensions: First, BMI was commenced in 2006. Second, after commencing, the variable has high rates of missing values. We did not find any change in BMI over time. This was unexpected considering previous knowledge on rising BMI in populations.[51] Selection bias and high missing rates may explain this. However, it is likely that BMI values should be accurate as measurements were made repeatedly during pregnancy. Therefore, the effect of measured BMI on other variables should be reliable.

The CROWN initiative suggested a set of core outcomes when conducting research on WWE in pregnancy and childbirth. [35] Maternal outcomes not included in our study but suggested by the CROWN initiative, were either not properly addressed in the MBRN or underpowered for inclusion in our study.

## Conclusions

Maternal complications in pregnancy and childbirth for WWE, developed similarly to that in women without epilepsy from 1999 to 2016. Only the risk of severe preeclampsia increased over time relative to women without epilepsy. Thus, neither the increased awareness of WWE as a group with special needs nor the change in pattern of AED use lead to a reduction in complications relative to women without epilepsy. There was a trend for a transition from elective cesarean section to induction of labour, specific for WWE with lamotrigine treatment. The increase in iatrogenic interventions for WWE indicate the need for continuous attention and follow-up. Reduction in smoking and increased use of folic acid supplements in WWE were positive factors, but the health care of WWE may still have room for improvement and benefit from individual risk assessment and active preventive measures.

## Supporting information

**S1 Fig. Total numbers of women with epilepsy using the four most common antiepileptic drugs in monotherapy during 1999–2016.**
(DOCX)

**S1 Table. Total complications in first pregnancies of women with epilepsy who used the four most common antiepileptic drugs in monotherapy compared to women without epilepsy 1999–2016.**
(DOCX)

**S2 Table. Maternal characteristics of 426 347 first births for women with epilepsy and for women without epilepsy 1999–2016.**
(DOCX)

## Acknowledgments

We are grateful for data provision by Medical Birth Registry of Norway. The Norwegian Epilepsy Association provided input to the research questions in this study.

## Author Contributions

**Conceptualization:** Kim Christian Danielsson, Nils Erik Gilhus, Ingrid Borthen, Rolv Terje Lie, Nils-Halvdan Morken.

**Data curation:** Kim Christian Danielsson, Rolv Terje Lie, Nils-Halvdan Morken.

**Formal analysis:** Kim Christian Danielsson.

**Investigation:** Kim Christian Danielsson, Nils-Halvdan Morken.

**Methodology:** Kim Christian Danielsson, Rolv Terje Lie, Nils-Halvdan Morken.

**Resources:** Kim Christian Danielsson.

**Supervision:** Ingrid Borthen.

**Visualization:** Kim Christian Danielsson.

**Writing – original draft:** Kim Christian Danielsson.

**Writing – review & editing:** Kim Christian Danielsson, Nils Erik Gilhus, Ingrid Borthen, Rolv Terje Lie, Nils-Halvdan Morken.

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
