## [Decision Letter · Decision Letter 0]

19 Aug 2019

PONE-D-19-16520

Maternal complications in pregnancy and childbirth for women with epilepsy: time trends in a nationwide cohort

PLOS ONE

Dear Dr Danielsson,

Thank you for submitting your manuscript to PLOS ONE. After careful consideration, we feel that it has merit but does not fully meet PLOS ONE’s publication criteria as it currently stands. Therefore, we invite you to submit a revised version of the manuscript that addresses the points raised during the review process.

We would appreciate receiving your revised manuscript by Oct 03 2019 11:59PM. To enhance the reproducibility of your results, we recommend that if applicable you deposit your laboratory protocols in protocols.io, where a protocol can be assigned its own identifier (DOI) such that it can be cited independently in the future. For instructions see: http://journals.plos.org/plosone/s/submission-guidelines#loc-laboratory-protocols

We look forward to receiving your revised manuscript.

Kind regards,

Angela Lupattelli, PhD

Academic Editor

PLOS ONE

**Journal Requirements**

**Comments to the Author**

1. Is the manuscript technically sound, and do the data support the conclusions?

Reviewer #1: No

Reviewer #2: Yes

2. Has the statistical analysis been performed appropriately and rigorously? 

Reviewer #1: No

Reviewer #2: Yes

3. Have the authors made all data underlying the findings in their manuscript fully available?

Reviewer #1: Yes

Reviewer #2: Yes

4. Is the manuscript presented in an intelligible fashion and written in standard English?

Reviewer #1: No

Reviewer #2: Yes

5. Review Comments to the Author

Reviewer #1: This is a paper describing obstetric complications in women with epilepsy (WWE) relative to women without epilepsy during 1996-2016 in Norway. Authors have identified all first singleton pregnancies from the Medical Birth Registry of Norway (MBRN) and extracted information on maternal epilepsy and antiepileptic drug use from the MBRN records. There are interesting elements in this study, but I have a lot of major and minor concerns which I will try to summarize below:

My major concern is the chosen methodology. Authors report significant changes in maternal characteristics and study outcomes both in WWE and in reference population. Also, there has been fundamental change in the pattern of AED use. Still, year of delivery/triennial period of delivery is not included in the adjusted regression models. Not even in the sensitivity analyses. In my opinion, time-varying elements have not been adequately addressed. Authors should adjust their risk estimates for year of delivery, or they could even consider alternative approaches such as age-period-cohort analysis. Another limitation is the BMI which was not possible to account for. Authors did not find any change in mean BMI 2006-2016 which seem surprising as obesity has increased during the last decades in many western countries. Moreover, it does not guarantee that 1996 and 2016 are comparable even though there would not have been change between 2006 and 2016. BMI is an important factor in terms of obstetric complications and could be tackled e.g. by imputation. It is not clear why country of origin is not among adjusted covariates as there is significant difference in proportions among WWE and reference population. Also, the proportion of immigrants has increased dramatically over the study period for which it is also an important factor for time trend analysis.

Another, somewhat related, major issue is that stratification-based sensitivity analyses do not handle the fact that maternal characteristics were not stable over time. Authors conclude that increase in relative risk of severe pre-eclampsia for WWE was not associated with other covariates. Only when the combination of covariates was considered, relative risk (RR) of severe pre-eclampsia in WWE was reversed. Variables included in the sensitivity analyses are those associated with socioeconomic status and these typically cumulate to same individuals. Therefore, one might similarly argue that combination should be in the final model and draw the conclusion that the risk of severe pre-eclampsia is not increased. In my opinion, authors make rather strong interpretation on lines 244-245. There is some bizarre ad hoc analysis on lines 297-305. Induction and elective c-section before term is due to life-threatening circumstances either for the mother or for the child. Thus, this is selected and biased population. I do not understand what is the aim and rationale of this analysis.

Some of the results seem controversial, e.g. increased risk for severe pre-eclampsia but decreased risk for mild pre-eclampsia in WWE. Authors haven´t paid much attention to this in the discussion. They say that their study lacks data on seizure activity and that never AEDs are associated with higher seizure frequencies in pregnancy. Could there be misclassification of seizure and severe pre-eclampsia/eclampsia so that epileptic seizures are registered as eclampsia? Could this explain why we don´t observe the same reduction of severe pre-eclampsia in WWE than in the reference population? Could there be verification bias due to more intensive surveillance for WWE women? In general, is there any biological explanation why risk pattern in WWE would be different for severe and mild eclampsia? There are 11 different outcomes and four individual AEDs. It is very likely that some of the results are significant only by chance because of the multiple testing for which P-values have not been corrected for.

There is quite a lot of underlying assumptions/interpretations that are not written out but assumed to be self-explanatory which they are not. E.g. it is not clear what WWE were excluded from analyses (lines 134-135). Those that used named four AEDs as polytherapy? Does any use only include only these four or any AED as monotherapy? More ambiguous text is listed under the minor concerns.

Minor concerns:

-increased focus on women with epilepsy? L70-71 What do you mean by increased focus?

-The concept of validity on lines 129-132. Two dimensions of data quality are completeness and accuracy. It is not clear how valid should be interpreted in this context.

-Outcomes of WWE? L95 What outcomes?

-Influence WWE in a specific way? L101. How?

-sections L142.

- should be obvious that induction or elective surgery are not spontaneous preterm births on L143-144.

-Sentence “Other epilepsy core…” on line 162 if needed to address as a limitation should be in the discussion.

-BMI abbreviation on L169 already defined on L88.

-In Figure S1 total number of woman (observations) is a suboptimal measure for pattern of use as prevalence of WWE is not stable. Would be better to report per 100 / 1000 births to describe changes in prevalence of use.

-Figure 1 consider rearrange so that there would be same scale on y-axis on figures within one row. P-values for trend women without epilepsy should be given in the Figure/Footnotes as interpretation of RRs is dependent on the rate in reference population.

-Figure 3 should be figure 1, describing the setting in which study outcomes derive from. It is not necessary to give P-values for trend for WWE in Figure 3 either, could be in the footnote as for the reference group.

-Lines 335-338, I don´t get the message.

-Lines 373-378, I don´t get the message.

Reviewer #2: The authors have performed a descriptive study of time trends in the nature and proportion of Norwegian women with epilepsy experiencing complications in pregnancy and childbirth over an eighteen year period. They also look at trends in maternal characteristics. Comparisons are made to the general Norwegian population. Key variables are defined and diagnoses have been validated. The study provides a wealth of information regarding these women. I have only a few comments: Given the size of the population, some of the p values are an overestimate of the potential differences, particularly with comparisons with women without epilepsy. While implied by the comparisons with the general population, this is not a hypothesis testing study, and, since the entire population is used, not a sample, the differences in proportions is what you have regardless of the "p" value.

In figure 3, some of the graphs do not start at 0 for the abscissa giving a false impression of the degree of change to the casual reader. A break in the y axis line would seem appropriate.

It is my perception that a protective effect of folate for eclampsia has not been demonstrated in clinical trials. (line 342) Some additional discussion here may be appropriate. Has the dietary food chain been modified in Norway to enhance folate intake in the general population?

I would say that while there may be more room for improvement in WWE, preconceptional counseling is important for all women (line 384 and 435)

6. PLOS authors have the option to publish the peer review history of their article (what does this mean?). If published, this will include your full peer review and any attached files.

Reviewer #1: No

Reviewer #2: No

---

## [Author Response · Author response to Decision Letter 0]

30 Oct 2019

Please see attached point-by-point response.

---

## [Editor Report · Decision Letter 1]

4 Nov 2019

Maternal complications in pregnancy and childbirth for women with epilepsy: time trends in a nationwide cohort

PONE-D-19-16520R1

Dear Dr. Danielsson,

We are pleased to inform you that your manuscript has been judged scientifically suitable for publication and will be formally accepted for publication once it complies with all outstanding technical requirements.

With kind regards,

Angela Lupattelli, PhD

Academic Editor

PLOS ONE
---

## [Editor Report · Acceptance letter]

11 Nov 2019

PONE-D-19-16520R1 

Maternal complications in pregnancy and childbirth for women with epilepsy: time trends in a nationwide cohort 

Dear Dr. Danielsson:

I am pleased to inform you that your manuscript has been deemed suitable for publication in PLOS ONE. Congratulations! Your manuscript is now with our production department. 

With kind regards,

on behalf of

Dr. Angela Lupattelli 

Academic Editor

PLOS ONE